# Application Route and Immune Status of the Host Determine Safety and Oncolytic Activity of Oncolytic Coxsackievirus B3 Variant PD-H

**DOI:** 10.3390/v13101918

**Published:** 2021-09-24

**Authors:** Ahmet Hazini, Babette Dieringer, Karin Klingel, Markian Pryshliak, Anja Geisler, Dennis Kobelt, Ole Daberkow, Jens Kurreck, Sophie van Linthout, Henry Fechner

**Affiliations:** 1Department of Applied Biochemistry, Institute of Biotechnology, Technische Universität Berlin, 13355 Berlin, Germany; ahmet.hazini@oncology.ox.ac.uk (A.H.); babette.dieringer@tu-berlin.de (B.D.); markpry79@gmail.com (M.P.); a.geisler@tu-berlin.de (A.G.); jens.kurreck@tu-berlin.de (J.K.); 2Department of Oncology, University of Oxford, Oxford OX3 7DQ, UK; 3Cardiopathology, Institute for Pathology and Neuropathology, University Hospital Tuebingen, 72076 Tuebingen, Germany; karin.klingel@med.uni-tuebingen.de; 4EPO GmbH Berlin-Buch, Robert-Rössle Str. 10, 13125 Berlin, Germany; Dennis.Kobelt@epo-berlin.com (D.K.); Ole.Daberkow@epo-berlin.com (O.D.); 5Berlin Institute of Health Center for Regenerative Therapies (BCRT), Campus Virchow Klinikum (CVK), Charité—Universitätsmedizin Berlin, Föhrer Str. 15, 13353 Berlin, Germany; sophie.van-linthout@charite.de; 6German Center for Cardiovascular Research (DZHK), Partner Site Berlin—Charité, Oudenarder Straße 16, 13316 Berlin, Germany

**Keywords:** coxsackievirus B3, oncolytic virus, cancer, colorectal carcinoma, microRNAs, PD

## Abstract

The coxsackievirus B3 strain PD-0 has been proposed as a new oncolytic virus for the treatment of colorectal carcinoma. Here, we generated a cDNA clone of PD-0 and analyzed the virus PD-H, newly generated from this cDNA, in xenografted and syngenic models of colorectal cancer. Replication and cytotoxic assays revealed that PD-H replicated and lysed colorectal carcinoma cell lines in vitro as well as PD-0. Intratumoral injection of PD-H into subcutaneous DLD-1 tumors in nude mice resulted in strong inhibition of tumor growth and significantly prolonged the survival of the animals, but virus-induced systemic infection was observed in one of the six animals. In a syngenic mouse model of subcutaneously growing Colon-26 tumors, intratumoral administration of PD-H led to a significant reduction of tumor growth, the prolongation of animal survival, the prevention of tumor-induced cachexia, and the elevation of CD3^+^ and dendritic cells in the tumor microenvironment. No virus-induced side effects were observed. After intraperitoneal application, PD-H induced weak pancreatitis and myocarditis in immunocompetent mice. By equipping the virus with target sites of miR-375, which is specifically expressed in the pancreas, organ infections were prevented. Moreover, employment of this virus in a syngenic mouse model of CT-26 peritoneal carcinomatosis resulted in a significant reduction in tumor growth and an increase in animal survival. The results demonstrate that the immune status of the host, the route of virus application, and the engineering of the virus with target sites of suitable microRNAs are crucial for the use of PD-H as an oncolytic virus.

## 1. Introduction

Oncolytic virotherapy is a new treatment for cancer. This therapy is induced by oncolytic viruses (OV) which infect and replicate in cancer cells without harming normal cells [1]. Tumor destruction induced by OV can be traced back to two fundamental mechanisms. First, direct virus-induced tumor cell lysis which is a result of lytic virus infection of the tumor cells. This leads to a second mechanism, the release of pathogen-associated molecular patterns (PAMP), danger-associated molecular patterns (DAMP), cytokines, tumor-associated antigens (TAA) and neoantigens from the infected tumor cells, which induce the development of systemic innate and adaptive anticancer immune responses [2].

The potential to selectively kill cancer cells has been demonstrated for various viruses, among them coxsackievirus B3 (CVB3) [3,4,5]. CVB3 is a small nonenveloped virus of about 30 nm of the family of the *Picornaviridae*. The virus has a positive-sense (+) single-stranded RNA genome of about 7.5 kb. The viral genome is terminated at its 5′-end by a nontranslated region (5′-NTR) which contains an internal ribosomal entry side that mediates translation of the viral RNA. The 3′-end contains a 3′-NTR with a poly(A) signal. Between the NTRs, the CVB3 genome encodes for a single continuous polyprotein. After translation this protein is proteolytically processed into four structural proteins (VP1 to VP4) which form the viral shell and into three nonstructural precursors (2BC, 3AB, 3CD) and seven further processed proteins (2A-2C; 3A-3D) which are required for distinct functions in the viral life cycle, such as processing of the viral protein and replication of the viral genome [6].

In humans, CVB3 induces mild self-limiting illness with flu-like symptoms in adults [7]. However, under certain circumstances individuals can develop myocarditis [8,9], pancreatitis [10] and aseptic meningoencephalitis [11].

The CVB3 serotype includes many genetic variants. These were either isolated from infected patients or represent laboratory strains which were generated by genetic engineering or emerged through in vitro or in vivo adaptation [12,13,14,15]. All CVB3 variants (Nancy, 31-1-93, H3, 2035A and PD) analyzed so far for oncolytic activity, showed potent anticancer efficiency in vivo, as shown in xenografted models of lung, colorectal, breast and endometrial cancer [3,4,5,16] and in a syngenic models of lung cancer [3]. On the other hand, undesirable virus-induced side effects occurred after the treatment of tumor-bearing mice. In particular, the animals developed pancreatitis and myocarditis induced by CVB3 infection of the heart and pancreas. The severity of the disease depended on the CVB3 strain [5,16,17,18]. Several studies demonstrated that side effects induced by oncolytic CVB3 can be prevented by insertion of target sites of microRNAs (miR-TS) into the virus genome. In fact, CVB3-induced pancreatitis and myocarditis was ablated by equipping the viral genome with miR-TS of tissue-specific microRNAs, such as the pancreas-specific expressed miR-375 and heart specific miR-1 or pancreas-specific miR-217 and miR-1 [17,19] or by using miR-TS of tumor suppressor miR-34a [18] or miR-143 and miR-145 [20].

Our group has focused on the development of the CVB3 variant PD as oncolytic virus for the treatment of colorectal carcinomas. PD emerged spontaneously in the course of adaptation of the CVB3 Nancy strain for growth in human fibroblasts [21]. Later this virus was passaged in CHO-K1 cells, leading to the variant PD-0 [5]. Compared to other members of the CVB group, both PD and PD-0 have a unique receptor tropism. Besides the coxsackievirus and adenovirus receptor (CAR) and the decay-accelerating factor (DAF), representing the primary and the cellular coreceptor of CVB3, respectively, both viruses can infect cells via binding to *N*- and 6-*O* sulfated heparan sulfates [5,22,23]. This extension of tropism has been shown in human colorectal cancer cells, in which PD-0 was more cytotoxic than other CVB3 strains [5]. In vivo, the PD-0 showed a potent inhibition of the growth of colorectal cancer in immunocompromised mice xenografted with subcutaneous (s.c.) growing colorectal tumors after intratumoral (i.t.) virus application. No signs of a systemic viral infection were detected in five of the six animals treated with PD-0, but one of the animals suffered an infection of the pancreas and heart. This, however, could be traced back to a mutant of PD-0 emerging in this animal [5].

The ready appearance of pathogenic mutants from strain PD-0 compromises its safety and may rule out its therapeutic use. To increase its safety, we generated a cDNA clone of PD-0, thereby producing the new PD variant we have named PD-H. We show that PD-H has similar functionality as PD-0 in vitro. In vivo, PD-H showed potent oncolytic activity in both xenografted and syngenic models of colorectal cancer. The safety of the virus, however, depended on the immune status of the treated animals and the route of virus administration. Furthermore, we show that virus-induced side effects in vivo can be prevented by insertion of miR-TS of the miR-375 into the genome of PD-H.

## 2. Materials and Methods

### 2.1. Cell Lines

HeLa cells were cultured in Dulbecco’s modified Eagle’s medium (DMEM) (Gibco, Karlsruhe, Germany) supplemented with 5% fetal calf serum (FCS) and 1% penicillin-streptomycin. HEK293T cell line was cultured in DMEM High Glucose (Biowest, Darmstadt, Germany) supplemented with 10% FCS, 1% penicillin-streptomycin (P/S), 1% L-glutamine and 1mM Na-pyruvate. The human colorectal carcinoma cell line DLD1 and the murine colorectal cancer Colon-26 were grown in RPMI 1640 (Invitrogen, Karlsruhe, Germany) supplemented with 10% FCS, 1% P/S, 1% L-glutamine and 1mM Na-pyruvate. The murine colorectal cancer cell line CT-26Luc was grown in DMEM (Thermo Fisher Scientific, Waltham, MA, USA) supplemented with 10% FCS for cultivation for animal experiments and with RPMI (Invitrogen) supplemented with 10% FCS, 1% P/S, 2 mM glutamate, 1 mM pyruvate, 10 mM HEPES and 0.1 mM NEAA. CT-26Luc cells were generated by stable lentiviral transduction for expression of the Renilla luciferase.

### 2.2. CVB3 Strains

The CVB3 strain PD-0 is an attenuated CVB3 variant. PD-0 emerged after passaging the CVB3 strain PD (GenBank: AF231765.1) several times through CHO-K1 cells in our laboratory. The strain 31-1-93 is a highly pancreato-and cardiotropic CVB3 variant. It was propagated in HeLa cells. The strain 31-1-93 was a kindly gift of Michaela Schmidtke (Institute of Virology and Antiviral Therapy, University of Jena, Germany). The strain Nancy (ATCC VR30) is a pancreato- and cardiotrophic CVB3 strain [24]. Nancy was propagated in HeLa cells. The strain H3 is a highly pancreato- and cardiotrophic CVB3 strain. It was generated from the CVB3-H3 cDNA clone pBK-CMV-H3 (kindly supplied by Andreas Henke, Institute of Virology and Antiviral Therapy, University of Jena, Germany) after transfection of the plasmid into HEK293T cells. All viruses were stored at −80 °C until use.

### 2.3. Cloning Full Length of cDNA of PD-0

Viral RNA of PD-0 was isolated by using High Pure Viral RNA Kit, (Roche Diagnostics, Mannheim, Germany) according to manufacturer’s instructions. Afterwards, reverse transcription was performed with 1µg isolated RNA and reverse primer incorporating *Cla*I restriction site (CVB3-7381ClaI, 5′-GTATCGATTTTTTTTTTTTTTTCCGCACCGAATGCGGAGAATTTA-3′) using 200 U Superscript III reverse transcriptase (Invitrogen) and 20 U RNasin (Promega, Mannheim, Germany). After heating at 55 °C for 1 h, then at 70 °C for 15 min, 2 U RNase H was added to the reaction tube and incubated at 37 °C for 30 min to remove RNA residues complementary to cDNA. The synthesized first strand cDNA was used as a template for amplification of full length PD-0 genome. Full length genome of PD-0 was amplified using 2 µL first-strand cDNA, 0.5 µL Q5 Polymerase (New England Biolabs, Ipswich, MA, USA), reverse primer (CVB3-7381ClaI) and forward primer incorporating *Not*I restriction site (CVB3-1NotI, 5′-GGTGCGGCCGCTTAAAACAGCCTGTGGGTTG-3′). The amplification cycles were as follows: one cycle at 98 °C for 30 s followed by 35 cycles at 98 °C for 15 s, 61 °C for 15 s and 72 °C for 5:30 min and termination with 72 °C for 5 min. The PCR reaction was separated using a standard 1% agarose gel and the 7.4 kb length PCR product was extracted from the gel and purified using Gel Purification Kit (Qiagen, Hilden, Germany). PCR amplicon was inserted directly into the pJET1.2/blunt vector using Clonejet PCR cloning kit (Thermo Fisher Scientific, Waltham, MA, USA) according to manufacturer’s instructions. The resulting plasmid was named pJET-CVB3-PD-H.

### 2.4. Construction of PD-H-375TS cDNA Expression Plasmid

The plasmid pJET-CVB3-PD-H-375TS was constructed by insertion of two copies of the miR-375TS into the 3′UTR of the PD-H genome in the pJET-CVB3-PD-H 24 nucleotides downstream of the stop codon of the viral polyprotein. The target sites were generated by single PCR reaction using the miR-375TS sense (5′-ACAAAAGCGCTTCACGCGAGCCGAACGAACAAAAGATTGGCTTAACCCTACTTTGA-3′) and miR-375TS antisense (5′-GTGAAGCGCTTTTGTTCGTTCGGCTCGCGTGAAAATTATTTCAAATTGTCTCTAATC-3′) primers and pJET-CVB3-PD-H. The primers were designed using the online infusion primer designing tool (Takara Bio, Shiga, Japan) and cloning was done by In-Fusion HD Cloning Kit (Takara Bio) according to manufacturer’s instructions.

### 2.5. Generation of PD-H by Transfection of Viral cDNA-Containing Plasmid

CHO-K1 cells were seeded into 24 plates at 10^5^/well and transfected next day with 800 ng pJET-CVB3-PD when the cells reached confluence of about 70% using the transfection regent PEImax (Polysciences Europe GmbH, Hirschberg an der Bergstrasse, Germany).

### 2.6. Generation of PD-H and PD-H-375TS by Transfection of Transcribed Viral cDNA

The plasmids pJET-CVB3-PD-H and pJET-CVB3-PD-H-375TS were linearized with *Pme*I and purified using NucleoSpin Gel and PCR Clean-up Kit (Macherey-Nagel, Düren, Germany). Viral RNA was produced using the in vitro T7 transcription kit (Roboklon, Berlin, Germany) according to manufacturer’s instructions. Isolated RNA was treated with DNAseI to digest the plasmids and purified with the RNeasy Mini Kit (Qiagen). To produce PD-H 4 × 10^5^ CHO-K1 and to produce PD-H-375TS, 7 × 10^5^ HEK-293T cells were seeded into 6-well plates. Cells were transfected with 2.5 µg of transcribed viral RNA with PEImax (Polysciences Europe GmbH). Four h post transfection, medium was removed from the cells and fresh medium was added. Cells were subjected to 3 freeze and thaw cycles 48 h after transfection to lyse the cells. Cell debris was removed by centrifugation at 10,000× *g* for 3 min and plaque assays were done to determine the titer of the virus. Transfection-derived PD-H and PD-H-375TS were propagated in CHO-K1 cells to reach adequate virus titers for use in experiments. For in vivo studies, both viruses were purified and concentrated by ultracentrifugation with 30% sucrose gradients as described previously [25].

### 2.7. Growth Curves

HeLa (1.1 × 10^5^), DLD-1 (1 × 10^5^) and Colon 26 cells (1.2 × 10^5^) were seeded into 24-well plates. Twenty four hours later the cells were infected with PD-0 or PD-H at a multiplicity of infection (MOI) of 0.3 for 1 h. Afterwards the supernatant was removed, and the cells were washed with PBS. Two ml fresh medium was added, and cell plates were incubated at 37 °C and 5% CO_2_. Plaque assays were performed for virus titration by collecting 100 µL supernatant 4 h, 24 h, 48 h and 72 h post-infection.

### 2.8. Silencing of PD-H-375TS by miR-375

HEK293T cells were seeded and transfected the next day, when they reached a confluence of 60% with the control plasmids pCMV-miR-216a (Origene Technologies, Rockville, MD, USA), expressing the miR-216a or pCMV-miR-375 (Origene Technologies) expressing the miR-375, each with PEI Max transfection reagent. The medium was replaced 48 h after transfection and cells treated with PD-H-375TS (MOI of 0.1) for 30 min at 37 °C. Following removal of viral solutions, fresh medium was added. Twenty four hours after virus infection, cells were subjected to 3 freeze/thaw cycles and the cell lysate was centrifuged to remove cell debris. The supernatant was used for determination of virus titers by plaque assay.

### 2.9. Virus Plaque Assays

Virus plaque assays were carried out as described previously [17].

### 2.10. Cell Killing Assay

The cell killing assay was carried out as described previously [5]. Briefly, Colon-26 cells were seeded in 96-well plates and virus solution was added at multiplicities of infection (MOI) of 1, 10 and 100. After 30 min incubation at 37 °C and 5% CO_2_, virus-containing medium was removed, fresh medium was added and cells incubated for 24 and 72 h. The cells were fixed with 10% trichloroacetic acid (TCA) (Carl Roth, Karlsruhe, Germany), stained with 30 µL crystal violet solution (Carl Roth) and photographed.

### 2.11. Cell Viability

Cell viability was assessed using Cell Proliferation Kits (XTT) (Promega GmbH, Walldorf, Germany) according to the manufacturer’s instructions. Briefly, cells were seeded onto 96-well plates and were infected at an MOI of 1, 10 or 100. At the indicated time points, absorbance levels were measured using a V-650 Spectrophotometer (Jason Inc. Milwaukee, WI, USA). As a negative control, cells were treated with 5% Triton X-100 solution.

### 2.12. Histopathological Analysis

Tissues were fixed in 4% paraformaldehyde and embedded in paraffin. Five µm thick tissue sections were cut and stained with hematoxylin and eosin (H&E) to visualize and quantify cell destruction and inflammation.

### 2.13. In Vivo Colorectal Cancer Models

All animal experiments were performed in accordance with the principles of laboratory animal care and all German laws regarding animal protection and approved by the responsible local authorities (State Office of Health and Social Affairs, Berlin, Germany). Before treatment, mice used for in vivo experiments were randomly assigned to the experimental groups.

### 2.14. Xenografted Subcutaneous DLD-1 Cancer Mouse Model

DLD-1 cells (5 × 10^6^ cells) were inoculated s.c. into the right and left flanks of 6-week-old female Balb/C nude mice (Charles River, Sulzfeld, Germany). When the tumor size reached ~0.5 cm diameter, one of the tumors was injected with 3 × 10^6^ pfu PD-H in a total volume of 20 µL. Control animals were injected with PBS. Two and four days later, the injection was repeated into the same tumor, again using 3 × 10^6^ pfu of PD-H or PBS.

### 2.15. Syngenic Subcutaneous Colon-26 Cancer Mouse Model

Colon-26 cells (5 × 10^5^ cells) were s.c. inoculated into the right and left flank of 6-week-old female Balb/C mice (Charles River). One of the tumors was intratumorally (i.t) injected with 1 × 10^6^ pfu of PD-H (in a total volume of 20 µL) when the tumor size reached ~0.5 cm in diameter. Animals were sacrificed 2 days, 3 days or 7 days post infection (p.i.). In another experiment, 6-week-old female Balb/C mice were s.c. inoculated into the right flank with 5 × 10^5^ Colon-26 cells. The tumors were injected with 3 × 10^6^ pfu PD-H when their size reached ~0.5 cm in diameter and reinjected one and two days later using the same virus dose. Control animals were i.t. injected with PBS. Animals were sacrificed at day 14 after first virus injection and analyzed.

### 2.16. Syngenic CT-26 Cell Cancer Mouse Model of Peritoneal Carcinomatosis

CT-26Luc (3 × 10^5^ cells) were inoculated intraperitoneally (i.p.) to 6-week-old female Balb/c mice (Charles River). Three, four and five days later 1 × 10^7^ pfu of PD-H-375TS was injected i.p. into the mice. Control mice received i.p. injections of PBS. All animals were analyzed for luciferase expression by bioluminescence measurement on days 3 and 11 after tumor cell inoculation.

### 2.17. Determination of Toxicity of PD-H In Vivo after i.p. Virus Application

PD-H or PD-H-375TS (each 5 × 10^6^ pfu) were i.p. injected into 6-week-old female Balb/C mice on three consecutive days. The animals were investigated 7 days after the first virus injection.

### 2.18. In Vivo Luciferase Measurement and Imaging

Bioluminescence imaging was performed by using the NightOwl LB 981 system (Berthold Technologies, Bad Wildbad, Germany). For bioluminescence imaging, mice were anesthetized with isofluran (Abbott GmbH, Wiesbaden, Germany) and received 150 mg/kg D-luciferin (Biosynth, Staad, Switzerland) dissolved in sterile PBS i.p. Tumor growth was monitored and semi-quantified by WinLight (Berthold Technologies) and ImageJ 1.48 k (NIH, Bethesda, MD, USA).

### 2.19. Mononuclear Cell Isolation from Tumor

Tumor mononuclear cells (MNC) were isolated from Colon-26 tumor-bearing mice, 14 days after intratumoral injection with PBS or PD-H using the Neonatal Heart Dissociation Kit (Miltenyi Biotec, Bergisch Gladbach, Germany) and gentleMACS Octo Dissociator (Miltenyi Biotec), according to the manufacturer’s instructions.

### 2.20. Flow Cytometry Analysis

Flow cytometry analysis of tumor MNCs was performed using directly conjugated monoclonal mouse antibodies: anti-mouse F4/80 Pacific Blue (clone BM8), anti-mouse CD49b PerCP/Cy5.5 (clone DX5), anti-mouse CD11c FITC (clone N418), and anti-mouse CD3 APC/Cy7 (clone 17A2) (Biolegend, San Diego, CA, USA). Surface staining was performed according to the manufacturer’s instructions. For intracellular staining with IL-1β, the MNCs were fixed and permeabilized for 20 min at 4 °C with Cytoperm/Cytofix (BD Biosciences, Franklin Lakes, NJ, USA) and stained with anti-mouse IL-1β APC (Monoclonal Rat IgG2B Clone #166931) (R&D Systems, Minneapolis, MN, USA), antibodies in BD Perm/Wash solution according to the manufacturer’s instructions. Sample analysis was performed on a MACSQuant Analyzer (Miltenyi Biotec) and flow cytometry data were analyzed with FlowJo 8.7. software (FlowJo, LLC, New York, NY, USA).

### 2.21. Statistical Analysis

Statistical analysis was performed with Graph-Pad Prism 8.2 (GraphPad Software, Inc., La Jolla, CA, USA). Results are expressed as the mean ± SEM for each group. Statistical significance was determined by use of the two-tailed unpaired Student’s t-test for cell culture investigations and by use of the Mann–Whitney U-test for in vivo investigations. Survival curves were plotted according to the Kaplan–Meier method and statistical significance determined by the (log-rank-test). Differences were considered significant at *p* < 0.05.

## 3. Results

### 3.1. Construction of a Full-Length cDNA Clone of PD-0

To generate a cDNA clone of PD-0, the viral RNA was extracted from the PD-0 stock, reverse transcribed and amplified by PCR. The gel purified PCR fragment (Figure 1A, left) was cloned into the pJET1.2 vector by blunt end ligation downstream of a T7 promoter. Two bacterial clones grew on the agar plate. Control digestion revealed that the plasmid clone pJet-CVB3-PD-H contained a band of the expected size of around 7.4 kb of the PD-0 genome **(**Figure 1A, right) and sequencing confirmed that the viral cDNA was inserted in the correct orientation (Figure 1B). The sequencing of viral cDNA revealed twelve nucleotide substitutions compared to the previously submitted PD sequence (Genbank accession no: AF231765) (Appendix A). One nucleotide substitution was found in the 5′-UTR at nucleotide position 610 (Appendix A). A further 11 nucleotide substitutions were detected in the protein encoding region of PD-0. Seven of them were silent mutations. Four mutations located in VP4 (amino acid (aa) 47 [T→A]), in VP3 (aa 34 [T→M] and aa 237 [F→Y]) and in the 2C protein (aa 42 [G→E]) resulted in aa changes compared to the published sequence of PD. We also directly sequenced the 5′-UTR and the VP1 to VP4 encoding region of PD-0, before the cDNA was cloned. Compared to PD-0, the viral cDNA in pJet-CVB3-PD-H had also one nucleotide substitution at nucleotide position 881 leading to aa change in the VP4 [aa 47 [T→A] (Appendix A).

This shows that PD-0 consists of different viral genotypes, with PD-H representing one of them. The more numerous differences that we found between PD-0/PD-H and PD seem to be best explained by the fact that the PD, which was originally established in human fibroblasts [21] has been adapted to the viral producer cell line CHO-K1.

### 3.2. Generation and Functional Analysis of PD-H In Vitro

To generate PD-H from pJet-CVB3-PD-H, two widely used techniques [18,24,25,26,27,28] were compared: the transfection with the viral cDNA and the transfection with viral RNA, which was transcribed from the viral cDNA. CHO-K1 cells were used as the producer cell line for PD-H. Both transfection methods resulted in the propagation of infectious virus already by 12 h post infection. However, the virus titers were about 2 to 3 orders of magnitude higher in cells transfected with viral RNA than in cells transfected with the viral cDNA. Similar results were seen 72 h after transfection (Figure 2A). Because of the production of distinctly higher viral titers after transfection of viral RNA, this method was used subsequently as the standard procedure to generate PD-H.

Next, we investigated whether PD-H retained the specific phenotypic and functional features of PD-0. Both viruses showed small plaques and similar plaque morphology, but plaque size was significantly larger in PD-H than in PD-0 (Figure 2B). Single step growth curves in HeLa cells revealed similar growth kinetics of both viruses in HeLa cells, the human colorectal cancer cell line DLD-1 and the murine colorectal cancer cell line Colon-26 (Figure 2C). Both viruses also lysed DLD-1 cells and Colon-26 cells with similar efficiency, as determined by XTT cell viability assays (Figure 2D). Interestingly, whereas DLD-1 cells are also highly sensitive to the CVB3 strains 31-1-93, H3 and Nancy [5,17], Colon-26 cells were completely resistant to these strains of virus (Appendix A).

Taken together, these results show successful generation of PD-H from a cDNA clone of PD-0. Moreover, PD-H grew similarly and induced cytotoxicity similar to that of PD-0 in colorectal carcinoma cell lines.

### 3.3. Oncolytic Activity and Safety of PD-H In Vivo

#### 3.3.1. PD-H Reduces Growth of Colorectal DLD-1 Cell Tumors and Increases Survival in Xenografted Mice, but Mutated Variants Emerge in Individual Animals

To investigate the safety and oncolytic activity of PD-H in vivo, we established subcutaneous DLD-1 cell tumors on both flanks of nude mice. One tumor was injected with 3 × 10^6^ pfu PD-H when the tumors reached a size of ∼0.5 cm. Two and four days later PD-H was re-applied at the same dose of virus. The control groups were injected with PBS instead of virus. Treatment with PD-H resulted in significant reduction of growth of both the injected and noninjected tumors (Figure 3A,B) and led to significant extension of survival compared to the control group (Figure 3C). One animal (mouse #M1) in the PD-H group was sacrificed on day 13 after the tumors were established because of signs of virus-induced sickness and a pronounced loss of body weight. The other animals were sacrificed as tumor growth became too large to be consistent with animal welfare regulations.

Histological examination revealed no pathological alterations in the pancreas, heart and spleen of four of the six investigated mice infected with PD-H. Mouse #M1 showed completely destruction of the pancreas, focal damage and inflammation of the heart and rarefication of spleen follicles with immune cells (Appendix A) and a second mouse in the PD-H group showed partial destruction of the pancreas, but no damage or inflammation of the heart. The analysis of virus burden revealed abundant amounts of virus in the injected and not injected tumor of PD-H injected mice, with the highest titers found in mouse #M1. Mouse #M1 also had high virus titers in the heart, spleen and pancreas. Sequencing of the coding sequence of VP1 to VP4 of the virus isolated from mouse #M1’s heart revealed two aa substitutions compared to PD-H, at aa 26 [E→V] and 230 [K→E] in the VP1. All other mice showed only very low titers in the heart, whereas all other investigated organs were virus-free (Figure 3D).

These data demonstrate that PD-H significantly reduced the growth of colorectal tumors in mice xenografted with DLD-1 tumors. However, in one individual animal, mutated variants of PD-H emerged which induced severe pancreatitis and myocarditis.

#### 3.3.2. PD-H Reduces Growth of Colon-26 Tumors in a Syngenic Mouse Model, Increases Survival Time of the Animals and Modifies the Immune Cell Profile within the Tumor Microenvironment, without Inducing Virus-Induced Adverse Effects

To investigate the efficiency and safety of PD-H in immunocompetent mice, we established Colon-26 tumors on the left flank of Balb/C mice and infected the tumors when they reached a size of about ∼0.5 cm with 3 × 10^6^ pfu of PD-H each day for three consecutive days. Control mice were injected with PBS. Compared to control mice, treatment with PD-H led to significant inhibition of tumor growth (Figure 4A–C).

No virus-induced adverse side effects were observed, and survival of PD-H-treated mice was significantly extended compared to the group of control mice (Figure 4D). Histological examination at day 14 after the first virus injection showed no pathological alterations in the heart, pancreas (Figure 4E), spleen, liver and brain (results not shown) of PD-H and control mice. At the same time point, virus was recovered from the tumors of two of four PD-H-infected mice. However, the levels of virus were low (about 10^2^ pfu/g). The pancreas, heart, spleen, liver and brain were virus-free (Figure 4F).

To elucidate whether PD-H spreads to distant Colon-26 tumors, we established two subcutaneous Colon-26 tumors on both flanks of Balb/C mice, injected one tumor with 1 × 10^6^ pfu of PD-H and investigated the animals 2, 3 and 7 days later. In the injected tumors, abundant amounts of PD-H were detected at each time point, but the virus titers dropped from 10^5^ pfu/g on day 2 and 3 to 10^4^ pfu/g on day 7. In two of five animals, PD-H was recovered from the contralateral tumor, which had not been injected directly, though the virus levels were considerably lower than in the injected tumor. Histological examination of tissue samples from these animals confirmed the absence of pathological alterations in normal organs at all time points (Appendix A).

The oncolytic activity of viruses is determined by the virus-induced activation of anticancer immune responses [2]. Therefore, we next investigated the immune cell profile within the Colon-26 tumor mass after PD-H infection using the model described above with one subcutaneous Colon-26 tumor. Following isolation of immune cells from the tumor on day 14 after the first virus injection, the levels of T cells, dendritic cells, natural killer (NK) cells and macrophages were analyzed. We found that the percentage of CD3^+^ T cells and dendritic (CD11c^+^) cells (DC) were significantly increased from 42% to 64% and 8% to 30%, respectively, in PD-H-treated compared to control tumors. In contrast, the percentage of NK (CD49^+^) cells and macrophages (F4/80^+^) dropped dramatically to nearly undetectable levels in PD-H treated tumors, while the level of these immune cells was around 15% in the tumors of the control animals (Figure 4G).

Several studies reported that subcutaneous inoculation of the Colon-26 cell line causes development of cachexia in mice [29,30]. In agreement with this, we observed a strong and progressive reduction of body weight in the untreated Colon-26 tumor-bearing control group after day 18 post-tumor-cell inoculation. Three of four mice had to be sacrificed when their body weight dropped below 80% of initial body weight on day 26. In contrast, none of the animals in the PD-H group suffered a reduction in body weight below 80%. Moreover, body weight stabilized or even increased towards the end of the experiment (Figure 4H). Elevated levels of proinflammatory cytokines in the tumor mass have been shown to induce cachexia. Particularly, tumor-infiltrating macrophages expressing IL-1β were found to be the major actor in this pathological condition [29,31]. To elucidate whether PD-H administration led to a change in the level of IL-1β expressing immune cells in the TME, we performed flow cytometry analysis from harvested tumor samples. The number of IL-1β-expressing macrophages (IL-1β^+^F4/80^+^), NK (IL-1β^+^CD49^+^), DC (IL-1β^+^CD11c^+^) and CD3^+^ cells (IL-1β^+^CD3^+^) was significantly reduced in PD-H treated tumors, compared to control mice (Figure 4I). As there was an increase of CD3^+^ and CD11c^+^ cells (Figure 4G), but a reduction of IL1β^+^CD11c^+^ and IL1β^+^CD3^+^ cell (Figure 4I) in the TME of PD-H compared to control mice, IL-1β-expressing CD3^+^ and DC cells were specifically reduced in the TME of PD-H mice. The same can be said for IL-1β-expressing NK and macrophages after we had calculated the ratio of F4/80+ and CD49+ (Figure 4G) and IL-1β^+^F4/80^+^ and IL-1β^+^CD49^+^ (Figure 4I) cells in the TME of control and PD-H mice, respectively.

All together, these data demonstrate that PD-H inhibits tumor growth, extends animal survival and prevents the development of cachexia in a syngenic Colon-26 model of colorectal cancer. Importantly, this occurs without inducing virus-induced undesirable side effects. Moreover, PD-H treatment led to a change in the pattern of immune cell invasion into the TME. Cells involved in the adaptive immune response were increased, while cells involved in the innate immune response, as well as IL-1β-expressing immune cells, were reduced in the PD-H treated mice.

#### 3.3.3. miR-375-Regulated PD-H Inhibits Intraperitoneal Growth of Colorectal CT-26Luc Tumors in Mice

As described above, PD-H inhibits the growth of subcutaneously growing colorectal tumors after i.t. application in xenograft and syngenic colorectal cancer models. In a clinically more relevant model of peritoneal carcinomatosis, we next investigated whether the intra-abdominal growth of murine colorectal CT-26Luc tumors can be inhibited by systemic application of PD-H via the intraperitoneal (i.p.) application route. To analyze whether PD-H is safe in this model, in a preliminary experiment 5 × 10^6^ pfu PD-H were injected i.p. into Balb/C mice and the animals were investigated 7 days later. The treated mice showed weak pancreatitis and myocarditis (Appendix A), but PD-H could not be detected by plaque assay in the heart and by real time RT-PCR in the pancreas of the infected mice, indicating that it was rapidly cleared from both organs. Previously, we have shown that the insertion of miR-375TS into the genome of the highly pancreato- and cardiotropic CVB3 strain H3 prevents virus-induced pancreatitis and myocarditis after i.p. virus application into mice [25]. Thus, we developed PD-H-375TS (Figure 5A) containing two copies of miR-375TS in the 3′-UTR of PD-H genome in order to prevent infection of pancreas and heart and induction of pancreatitis and myocarditis, respectively.

To prove the sensitivity of the new PD-H-375 to miR-375, HEK293T cells were transfected with the miR-375 expression plasmid or a control plasmid expressing the miR-216a and infected with PD-H-375TS. As shown in Figure 5B, the replication of PD-H-375TS was about one order of magnitude lower in miR-375 than in miR-216 expressing cells, indicating that replication of PD-H-375TS was specifically inhibited by miR-375. Next, we compared replication of PD-H-375TS with parental PD-H in the murine colorectal cancer cell line CT-26Luc in vitro. Plaque titration 24 h after virus administration revealed that both viruses showed similar growth, indicating that equipping PD-H with miR-375TS did not affect replication of the virus in this target cell line (Figure 5C).

To prove the safety of PD-H-375TS in vivo, 5 × 10^6^ pfu of the virus were applied i.p. to Balb/C mice. The virus could not be detected in the pancreas, or the heart and no pathological alterations were observed in these organs (results not shown) on day 7 after virus administration. Thus, PD-H-375TS did not induce toxicity after i.p. application in vivo.

For examination of the oncolytic efficacy of PD-H-375TS, we established intra-abdominal growing CT-26Luc tumors by injection of CT-26Luc tumor cells into the abdominal cavity of Balb/C mice. Three, four and five days after tumor cell administration the animals were injected i.p. with 1 × 10^7^ pfu of PD-H-375TS. Bioluminescence imaging showed significantly lower CT-26luc tumor growth on day 11 after tumor cell implantation in PD-H-375TS group compared to the control group, whereas there was no difference between both groups on day 3 after tumor cell inoculation (Figure 5D). In accordance with reduced tumor growth, the survival of PD-H-375TS treated mice was significantly extended in PD-H-375TS-infected mice compared to the control mice (Figure 5E). To assess replication of PD-H-375TS in normal tissue, the pancreas and the heart of infected mice were investigated on the day when animals were sacrificed. Using real-time RT-PCR, PD-H-375TS genomes were detected in the pancreas of one of six infected mice, but at very low levels of 1 × 10^2^ virus genome copies per µg tissue, whereas the hearts of all PD-H-375TS-infected animals were virus-free. There were no pathological findings in either organ, as noted by histopathological examinations.

These data demonstrate that PD-H-375TS is able to inhibit intra-abdominal growth of murine colorectal tumors and does not induce virus-induced side effects after i.p. virus application.

## 4. Discussion

Over the last few years, the oncolytic activity of different strains of the CVB3 has been demonstrated for the experimental treatment of cancer, among them, the CVB3 strain PD-0 [3,4,5,16]. The ability of PD-0 to use different cellular receptors to enter tumor cells is thought to be the reason for its broad tumor cell tropism and its high oncolytic activity. In addition, PD-0 is severely attenuated in vivo, although little is known about the underlying mechanisms [5,32]. However, in a previous work we found that after i.t. administration of PD-0 to DLD-1 tumor-bearing nude mice, a single animal developed severe pancreatitis and myocarditis [5], which represents the typical pattern of organ infection observed in mice after infection with pathogenic CVB3 strains [12]. Detection of an aa substitution in VP1 and two aa substitutions in VP2 compared to PD-0 in the affected mice demonstrated that the disease was induced by a PD-0 mutant [5].

Because of the frequent misintegration of nucleotides within the viral genome [33] and the virus’ ability to generate chimeras by genomic recombination [34], the viral population of a picornavirus invariably contains a large number of genotypic variants, which enable the virus to adapt rapidly to a new environment. The dynamics of the process ensure that the number of mutations increases as the virus is passaged [35]. Based on these facts, we hypothesized that use of a cDNA clone of PD-0, together with an optimized protocol of virus amplification, would minimize the accumulation of mutants within the virus solution, so that its application to tumor-bearing mice would prevent, or at least decrease the frequency of the appearance of pathogenic variants of the virus in vivo. We successfully cloned a full-length infectious cDNA of PD-0 and showed that the variant PD-H generated by this clone is as efficient as PD-0 in vitro and in vivo. However, unfortunately, after i.t. injection into DLD-1 tumor xenografts, one mouse became visibly sick and had high virus titers in its pancreas, heart and spleen. Histological results confirmed pancreatitis and myocarditis in this mouse. Very similarly to our previous study, these results were triggered by a mutant of the applied virus. Thus, our approach to prevent the occurrence of pathogenic PD-H mutants by use of a cDNA clone failed. Nevertheless, it remains a mystery why pathogenic PD-0 and PD-H mutants emerged in individual treated mice. Comparable results have never been reported for other oncolytic CVB3 strains used in xenografted cancer models in mice [3,4,16]. The adaptation of viruses is a process which aims at optimizing viral replication in a suboptimal host environment [36]. It is a process that is directed and affected by the mutation rate of the virus and the specific cellular environment [37,38]. PD-0 and PD-H mutants showed only a few mutations, compared to their parental strain PD-0 and PD-H, respectively. Most of them were found in the capsid proteins, suggesting that the emergence of pathogenic mutants may be related to the changes in virus-receptor interactions and virus uptake. Interestingly, one aa substitution at aa 230 [K→E] in the VP1 was found in both mutants. Hence this mutation may be of specific importance for the pathogenic phenotype of both mutants. This observation also indicates that when used in the same living system (DLD-1 tumor-bearing mouse), the virus evolution may be similar. Conversely, this conclusion raises the question of whether another environment would prevent the development of PD-0/PD-H mutants. Indeed, we did not find PD-H mutants in Colon-26 and CT-26 tumor mouse models. Animals used in CT-26Luc and Colon-26 cancer models had an intact immune system with an efficient T cell immune response that is lacking in nude mice [39]. It enables a more efficient clearance of virus and less severe illness, compared to immunodeficient mice [40,41,42]. Conceivably, virus mutants may be cleared before they manage to spread within the host.

Because of the uncertainty regarding whether PD-H mutants may emerge in cancer patients, and our observation that PD-H can induce weak pancreatitis and myocarditis after i.p. application, further improvements of PD-H are necessary. The sensitization of the oncolytic CVB3 to tumor suppressor- or organ-specific microRNAs has been proven as an effective approach to prevent their replication in natural target organs [17,18,19,20,25,43]. Accordingly, when we equipped PD-H with target sites of miR-375TS, the virus PD-H-375TS was unable to infect and damage the pancreas and the heart of mice after i.p. injection, even when it was applied at a high dose.

The safety and efficiency of oncolytic CVB3 have been reported in different xenografted mouse models of cancer [3,4,5,16,18], but until now, the oncolytic activity of CVB3 has only been confirmed in one syngenic tumor model, in which subcutaneous murine TC-1 lung cell tumors were injected with the CVB3 Nancy strain [3,18]. Here, we report that PD-H significantly inhibits tumor growth and extends the time of survival of the animals in a syngenic mouse model of an s.c.-growing colorectal Colon-26 cell tumor. Moreover, similar results were observed after systemic administration of PD-H-375TS in a syngenic model of peritoneal carcinomatosis induced by colorectal CT-26Luc-tumor cells. Importantly, in both models there were no adverse virus-induced effects detected after treatment with the respective viruses, demonstrating that the PD-H and PD-H-375 are safe for the respective applications. As previously documented in immunocompromised mice [3,4,5,16], we also observed that the virus spreads from the primary infection to a distant tumor in immunocompetent mice. Following infection in primary target organs, CVB3 is able to access secondary target organs via the blood stream [25]. It is not known whether this occurs exclusively via free-circulating virus or whether virus-infected immune cells also play a role. The latter may be possible for PD-H moving to distant tumors, as we observed that PD-0 can infect immune cells [5]. In the distant tumor, the virus may destroy tumor cells as in the primary tumor and promote antitumor immune responses, leading to the destruction of the distant tumor. However, two studies using oncolytic Newcastle disease virus and the herpes simplex virus type 1 mutant G207 have shown that a distant tumor can be destroyed by OV, although it is not infected [44,45]. The systemic innate and adaptive tumor-specific immune response, induced by the release of tumor-specific antigens and neo-antigens from the primary infected and lysed tumor and directed against these tumor-specific proteins displayed on uninfected cancer cells is thought to be the main trigger in this process [46,47,48]. The same mechanisms may also contribute to distant tumor destruction by oncolytic CVB3. Whether infection of the distant tumor may increase CVB3-induced oncolysis is not known and needs to be elucidated in further investigations.

The induction of immunogenic cell death and the release of tumor-specific antigens, neoantigens, PAMPS and DAMPS into the TME contributes to the activation of antigen-presenting cells as DC. These cells migrate from the TME to the lymph nodes where they mature and prime T cells and induce a cancer-specific T-cell response [2]. Miyamoto et al. [3] showed that CVB3 can induce immunogenic cell death and treatment of lung tumors in nude mice leading to changes in the TME, as shown by a rapid increase in the levels of granulocytes, DC and NK cells after virus administration. We found that treatment of immunocompetent mice bearing Colon-26 tumors with PD-H resulted in very strong decrease of NK cells and macrophages in the TME. Since we carried out the measurement on day 14 after virus administration and Miyamoto et al. measured on day 2, this may be the main reason for the observed differences. NK cells and macrophages are parts of the innate immune system and are the first wave of immune cells, which infiltrate into the virus-infected tumor [2] in both immunocompromised and immunocompetent mice. In immunocompetent hosts, the invasion of T cells supports the defense. This was confirmed in this study. The levels of CD3^+^ and CD11c^+^ were significantly elevated in the TME of Colon-26 tumors. Hence, also the second arm of the host immune response, the adaptive immune response, is activated by CVB3 and similar to other OVs, this may be crucial for the oncolytic efficiency of PD-H.

The Colon-26 cancer model has been used for a long time as the standard model to study cancer cachexia [30,31], a disease which is characterized by strong body weight loss that is associated with the progressive loss of skeletal muscle mass and adipose tissue and a severely disturbed metabolism [49]. Treatment with PD-H stopped the progress of initial body weight loss of Colon-26 tumor-bearing mice and even reversed it, whereas control mice developed cachexia and had to be sacrificed. Thus, our data support previous studies with adenoviruses and vaccinia viruses showing that OV can prevent the development of cancer cachexia [50,51,52]. Proinflammatory cytokines with catabolic actions, in particular, IL-6, IL-1β, TNFα and IFNγ, seem to be key mediators of cancer cachexia and it has been proposed that crosstalk between tumor cells, the immune system and the specific cell composition of the TME are crucial to the releasing of these factors [53,54]. In line with this, we observed a strong reduction of IL-1β expressing macrophages, NK cells, DC and CD3^+^ cells in the TME of PD-H-infected Colon-26 tumors demonstrating that the immunomodulating properties of PD-H contributed to the reduction of IL-1β-expressing immune cells, which may also affect cachexia development. However, it should be noted, that cachexia development is complex. Thus, further investigations are necessary to elucidate the mechanisms involved in PD-H-induced prevention of cancer cachexia.

In conclusion, here we successfully generated a cDNA clone of the CVB3 variant PD-0 and from this clone the recombinant CVB3 variant PD-H. Our data demonstrate that the safety of the PD-H in vivo depends on the immune status of mice, the application route of the virus and the genetic engineering of the virus with miR-375TS. In immunocompetent mice, PD-H had potent oncolytic activity against colorectal carcinomas and can prevent the development of tumor cachexia. The induction of an adaptive systemic anticancer immune response represents a crucial mechanism in the process. PD-H as well as the PD-H-375 may be suitable for use in immunocompetent hosts.

## Figures and Tables

**Figure 1 viruses-13-01918-f001:**
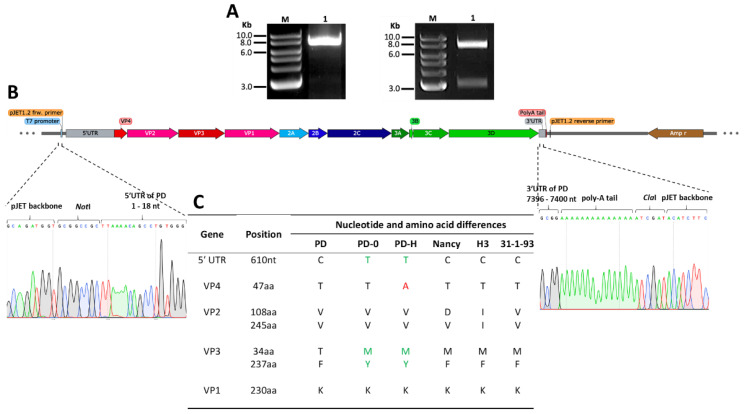
Construction of cDNA plasmid clone pJet-CVB3-PD-H. (**A**) Viral RNA isolated from PD-0 was reverse transcribed and full-length cDNA was amplified by PCR. Left image: a ∼7.5 kb long final product was confirmed on agarose gel. Lane M; 1-log DNA ladder, Lane 1; amplified cDNA from PD-0. Right image: cDNA from PD-0 was extracted from the gel and inserted into the pJET1.2/blunt cloning vector. The resulting plasmid pJET-CVB3-PD-H was digested with *Not*I and *Cla*I restriction enzymes which flank the viral cDNA. Lane M; 1-log DNA ladder, Lane 1 upper band; full length cDNA of PD-0, lower band; vector backbone. (**B**) Schematic presentation of viral cDNA within pJet-CVB3-PD-H. Chromatograms obtained from Sanger sequencing confirmed the correct orientation of the viral cDNA in the plasmid pJET1.2 backbone. Left sequence chromatogram represents the junction of the pJET1.2 backbone and cDNA of PD-0 with the first 17 nucleotides of the 5′-UTR region of the viral cDNA. Right chromatogram represents the junction of the pJET1.2 backbone with the 3′ terminus of the 3′-UTR region of cDNA of PD-0 with the poly-A tail. (**C**) Nucleotide sequence comparison of the viral 5‘-UTR and aa sequence comparison of the viral proteins VP1 to VP4 of different CVB3 variants. Note, the cDNA clone of PD-0 is named PD-H. PD (Genbank accession number: AF231765), 31-1-93 (Genbank accession number: AF231763), Nancy (Genbank accession number: M16572.1), H3 (Genbank accession number: U57056). Green letters: differences between PD-0 and PD-H and other CVB3 variants. Red letters: Difference in aa sequence between PD-0 and PD-H.

**Figure 2 viruses-13-01918-f002:**
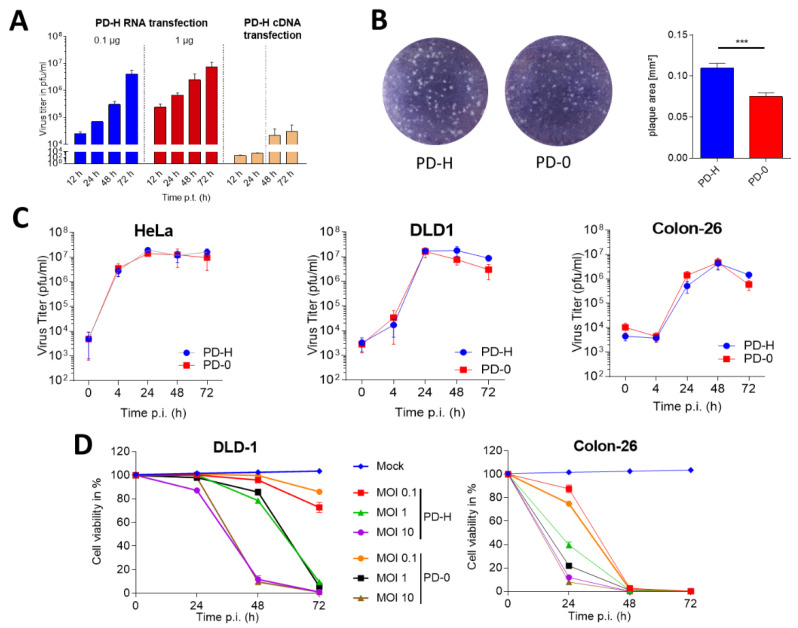
Optimization of production of PD-H and comparison of virus function of PD-H and PD-0. (**A**) CHO-K1 cells were seeded in 6-well plates and transfected with 0.1 or 1 µg of in vitro transcribed genomic RNA of PD-0 cDNA or with 800 ng pJET-CVB3-PD. The virus titers were determined by plaque assay in HeLa cells at the indicated points in time. (**B**) Plaque morphology and size. Plaques of indicated viruses were determined by plaque assays on HeLa cell monolayers. Left panel: representative images of viral plaques (white dots). Right diagram: plaque diameter of PD-H and PD-0 of 50 counted plaques. Significance: *** *p* < 0.001. (**C**) Virus growth kinetics. HeLa, DLD-1 and Colon 26 cells seeded in 24-well plates and infected with 0.3 MOI PD-H or PD-0. The amount of virus progeny in whole cell lysates was measured by plaque assay on HeLa cells at indicated time points. (**D**) Cell viability. DLD-1 and Colon 26 cells seeded in 96-well plates were infected with PD-H or PD-0 at the indicated MOIs. Cell viability was assessed 24 h, 48 h and 72 h after infection. Data in diagrams are show as mean values ± SEM.

**Figure 3 viruses-13-01918-f003:**
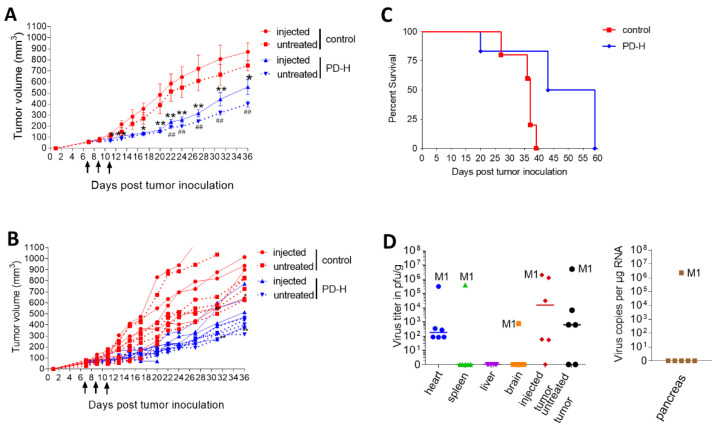
Oncolytic activity and safety of PD-H in nude mice with DLD-1 cell tumors. DLD-1 cell tumors were established on both flanks of Balb/C nude mice. When the tumor size reached ~0.5 cm diameter, one of the tumors was injected with 3 × 10^6^ pfu PD-H (*n* = 6) or PBS (control, *n* = 5), while the contralateral tumor remained untreated. Two and four days after injection, the same tumor was reinjected using each 3 × 10^6^ pfu of PD-H. (**A**) Tumor growth. The data are shown as mean values ± SEM for each group. Injected means the tumor which was injected with PBS (control) or with PD-H; untreated means the contralateral tumor of the same animal which was not injected. Significance between control vs. PD-H group: injected tumor * *p* < 0.05; ** *p* < 0.01; untreated tumor ^##^
*p* < 0.01 (**B**) Tumor growth. Data of A shown for each animal. (**C**) Kaplan−Meier survival curve. Significance between control and PD-H-treated mice; *p* = 0.0179. (**D**) Biodistribution of PD-H. Organ samples from mice were collected on the day when the animals were sacrificed. M1 represents data points of mouse #M1, which was sacrificed on day 13 after tumor cell administration because of signs of virus-induced sickness and pronounced loss of body weight. Left diagram: PD-H titers determined by plaque assay in indicated organs and in the injected and untreated contralateral tumors. Right diagram: Virus genome copy number in the pancreas as determined by qRT-PCR. Note: pancreatic tissue enzymes make it impossible to perform proper analysis of virus titer using plaque assays as pancreatic tissue lysate causes cytotoxicity in HeLa cells, which we used for the plaque assays. Therefore, we always perform quantitative RT-PCR to measure presence of viral RNA in pancreatic tissue. Data are shown as medians for each group and for each animal.

**Figure 4 viruses-13-01918-f004:**
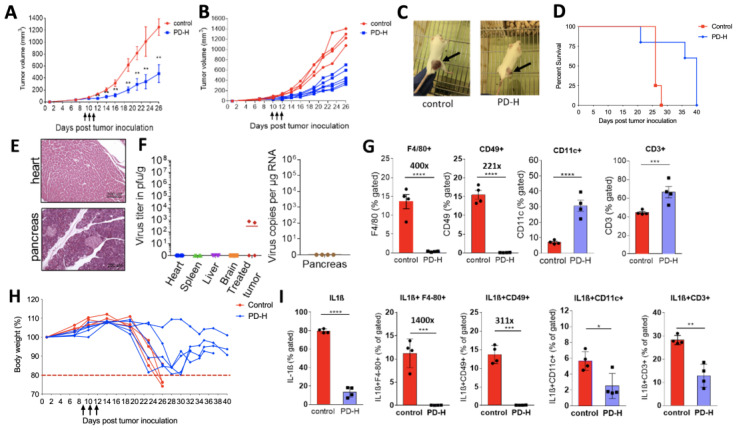
Oncolytic efficiency of PD-H in a Colon-26 tumor syngenic mouse model. Colon-26 cells were subcutaneously inoculated onto the right flank of BALB/c mice. When the tumor size reached ~0.5 cm diameter 3 × 10^6^ pfu of PD-H (*n* = 6) or PBS (control, *n* = 4) was injected i.t. Injection of 3 × 10^6^ pfu PD-H or PBS by the same route was repeated after one and two days. (**A**) Tumor volumes are shown as means ± SEM for each group. Significance: ** *p* < 0.01. (**B**) Data of A shown for each animal. (**C**) Image of PD-H-treated Colon-26 tumor mice. PD-H treated mouse: black arrow shows PD-H injected tumor; control mouse: black arrow shows PBS-injected tumor. Images were taken on day 20 after tumor cell injection. (**D**) Kaplan–Meier survival curve. Significance: *p* = 0.0027. (**E**) Histological examination of heart and pancreas. Shown are representative tissue slides after staining with H&E on day 14 after first PD-H injection. (**F**) Biodistribution of PD-H. Virus titers were determined as described under 3D. Investigations were carried out on day 14 after the first PD-H injection. (**G**) Change of immune cell infiltration in the tumor microenvironment (TME) by PD-H. Colon-26 tumors were harvested 14 days after the first injection with PD-H or PBS. Shown are the gated percent of T cells (CD3^+^), DC cells (CD11c), NK cells (CD49) and macrophages (F4/80). Significance: *** *p* < 0.001, **** *p* < 0.0001. (**H**) Body weight. Left diagram: body weight for each animal. The dashed red line at 80% shows the threshold below which the animals have to be sacrificed due to animal welfare guidelines. All PD-H treated animals were sacrificed when the tumor size reached upper limit of 1.8 cm^3^. (**I**) PD-H treatment decreased the level of IL-1β expression in the tumor microenvironment. Tumors were harvested 14 days after first PD-H or PBS injection. Shown are gated percent of IL-1β expressing T cells (IL-1β^+^CD3^+^), dendritic cells (IL-1β^+^CD11c^+^) macrophages (IL-1β^+^F4/80^+^) and NK cells (IL-1β^+^CD49^+^). Significance: * *p* < 0.05, ** *p* < 0.01, *** *p* < 0.001, **** *p* < 0.0001.

**Figure 5 viruses-13-01918-f005:**
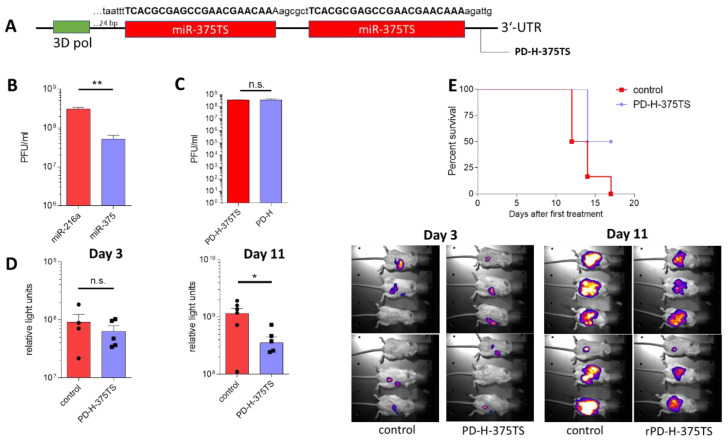
Oncolytic activity of PD-H-375 against intra-abdominal growing CT-26 tumors. (**A**) Scheme of miR-375TS region within the genome of PD-H-375TS. Two copies of the miR-375TS, separated by a 6 nucleotide long spacer, were inserted into the 3′-UTR of PD-H 24 bp downstream of the 3D polymerase encoding sequence. The miR-375TS are 100% complementary to the pancreas-specific expressed miR-375. The sequence of miR-375TS is shown in capitals. (**B**) Silencing of PD-H-375TS by miR-375. HEK293T cells were transfected with pCMV-miR-216a or pCMV-miR-375 and infected 48 h later with 0.1 MOI PD-H-375TS. Twenty-four hours later virus titers were determined by plaque assay. Shown are mean values ± SEM from two experiments performed, each in triplicate. Significance: ** *p* < 0.01. (**C**) Comparison of PD-H-375TS and PD-H replication in CT-26Luc cells. HEK293T cells were seeded in 24-well plates and infected with PD-H or PD-H-375TS at an MOI of 1. Twenty four hours later the viruses were isolated and the virus titers determined by plaque assay. (**D**) Oncolytic activity of CVB3-PD-375TS in tumor model of i.p. growing CT-26Luc tumors. The 3 × 10^5^ stable luciferase-expressing CT-26Luc cells were i,p. injected into Balb/C mice. At day 3, 4, and 5 after CT-26 tumor cell injection each 1 × 10^7^ pfu PD-H-375TS (*n* = 6) or PBS (control) were i.p, injected to mice. Luciferase expression was determined by bioluminescence imaging. The diagrams on the left show the luciferase activity on days 3 and 10 after CT-26Luc inoculation. Right: images of animals after application of luciferase at day 3 and 10 after tumor cell injection. * *p* < 0.05. (**E**) Kaplan–Meier survival curve. Significance: *p* = 0.0266.

## Data Availability

Data will be supplied following reasonable requests.

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
