# Peer review of "Application Route and Immune Status of the Host Determine Safety and Oncolytic Activity of Oncolytic Coxsackievirus B3 Variant PD-H"

_viruses, 2021, doi:10.3390/v13101918_

Round 1
Reviewer 1 Report
I think this manuscript is well organized manuscript. minor suggestions:
- Line 288, remove 12 or twelve
- What is the volume for intratumoral injection?
- Figure 3: According to the legends, A,B,C were from the same experiment. Since C has a duration of more than 50 days, readers would probably like to see the tumor volume figure (A, B) extends to the same endpoint day. It appears control mice all euthanized before around 33, but in A, B, some mice still alive in control group, error or any explanation? Why don’t include titer of pancreas in the same figure of other organs? If the pancreas tissue is not enough for titration, readers would probably like to see the viral RNA load of tumor and heart in the same figure of pancreas if applicable. If not the same experiment, would like to see annotation in the legends.
- Figure 4: As mentioned previously, recommend same endpoint day in tumor volume, survival and body weight figure if applicable. Besides, it appears the cachexia caused weight loss were successfully reversed in the treatment group, then I suppose the tumor disappeared. Thus, it would be great to know if the improvement of cachexia caused body weight loss associated with tumor shrinking. If not the same experiment, would like to see annotation in the legends.
- Line 601-602, reverse mutation also observed in reference 20.
- Line 671: by or be?
- Line 683, about the discussion of cachexia and the consequent weight loss, I am wondering if data on day 26-28 and day 38 (according to figure 4H) of the immune marker data (e.g. figure 4I on day 14) is available, as on day 14 (Fig 4I), there’s no difference between two groups.
Author Response
Reviewer: #1
- Line 288, remove 12 or twelve
Answer: We appreciate the reviewer’s suggestion, 12 has been removed.
- What is the volume for intratumoral injection?
Answer: The volume was 20µl for intratumoral injection. We added this in the material and methods (line 225 and line 231).
- Figure 3: According to the legends, A,B,C were from the same experiment. Since C has a duration of more than 50 days, readers would probably like to see the tumor volume figure (A, B) extends to the same endpoint day. It appears control mice all euthanized before around 33, but in A, B, some mice still alive in control group, error or any explanation?
Answer: While the starting timepoint in figures A and B were “days post tumor inoculation”, in Figure 3C the starting time point was after the first dose of treatment. As this might be the reason of confusion, we have changed the figure 3C by rearranging the starting date from tumor inoculation similar to figure 3A and 3B. As we had to sacrifice second control mouse on day 36 post tumor inoculation, three mice left after that time point in this study group. This number was not enough to make proper statistical significance, therefore after this time point, we stopped to record tumor size daily. Due to statistical analysis, demonstration of tumor size with the earlier endpoint compared to the survival rate can be seen frequently in such studies (e.g. Reference 20).
- Why don’t include titer of pancreas in the same figure of other organs? If the pancreas tissue is not enough for titration, readers would probably like to see the viral RNA load of tumor and heart in the same figure of pancreas if applicable. If not the same experiment, would like to see annotation in the legends.
Answer: Pancreatic tissue enzymes make impossible to perform proper analysis of virus titer using plaque assays as pancreatic tissue lysate causes cytotoxicity in HeLa cells, which we used for the plaque assays. Therefore, we always perform quantitative RT-PCR to measure presence of viral RNA in pancreatic tissue. On the other hand, we use plaque assay to measure viral titer in all other tissue samples as it is cost-effective and needs less manpower compared to the quantitative RT-PCR. To explain the reader why we used RT-PCR for determination of virus titers in the pancreas we inserted a note in the legend of figure 3 (line 393-396).
- Figure 4: As mentioned previously, recommend same endpoint day in tumor volume, survival and body weight figure if applicable. Besides, it appears the cachexia caused weight loss were successfully reversed in the treatment group, then I suppose the tumor disappeared. Thus, it would be great to know if the improvement of cachexia caused body weight loss associated with tumor shrinking. If not the same experiment, would like to see annotation in the legends.
Answer: Unfortunately, same endpoint day for tumor volume, survival curve and body weight are not applicable. In this experiment the endpoint day for the tumor volume was day 26, as we had to sacrifice 3 out of 4 control mice in that day. After day 26 demonstration of statistical analysis for tumor volume was not possible. We have rearranged the Figure 4D and 4H by changing the timeline from “days after first treatment” to “days after tumor inoculation” as was shown in Figure 4A and 4B. There was no direct correlation between the tumor size and cachexia as all PD-H treated mice were sacrificed due to tumor size. We have inserted a sentence in the legends.
- Line 601-602, reverse mutation also observed in reference 20.
Answer: We appreciate the reviewer’s suggestion, Lie et al. detected reverse mutation in inserted target site sequences, however in line 601-602 (now 606-607) we mention about occurrence of mutation in the viral genome.
- Line 671: by or be?
Answer: It is corrected. Thank you.
- Line 683, about the discussion of cachexia and the consequent weight loss, I am wondering if data on day 26-28 and day 38 (according to figure 4H) of the immune marker data (e.g. figure 4I on day 14) is available, as on day 14 (Fig 4I), there’s no difference between two groups
Answer: Unfortunately, we didn’t analyse immune cells in indicated time points. We are currently designing a large in vivo study to investigate cachexia extensively. Figure 4I data were obtained from another experiment and we would like to mention that in Figure 4I the date of the investigation was 14 days post first dose injection which is equal to 23 days post tumor inoculation (tumors were injected 8 days after tumor inoculation). This means in both studies we observed cachexia in animals at similar time points.
Reviewer 2 Report
Hazini A.et al, generated a cDNA clone of PD-0 and analyzed the newly from this 24 cDNA generated virus PD-H in xenografted and syngenic models of colorectal cancer. Replication and cytotoxic assays revealed that PD-H replicated and lysed colorectal carcinoma cell lines in vitro as well as PD-0. Intratumoral injection of PD-H into subcutaneous DLD-1 tumors in nude mice resulted in strong inhibition of tumor growth and significantly prolonged survival of the animals, but 28 virus-induced systemic infection was observed in one of the six animals. In a syngenic mouse model 29 of subcutaneously growing Colon-26 tumors, intratumoral administration of PD-H led to a significant reduction of tumor growth, the prolongation of animal survival, the prevention of tumor-induced cachexia, and elevation of CD3+ and dendritic cells in the tumor microenvironment. No virus-induced side effects were observed. After intraperitoneal application, PD-H induced weak pancreatitis and myocarditis in immunocompetent mice. By equipping the virus with target sites of 34 miR-375, which is specifically expressed in the pancreas, organ infections were prevented. Moreover, employment of this virus in a syngeneic mouse model of CT-26 peritoneal carcinomatosis resulted in significant reduction in tumor growth and an increase in animal survival. They concluded that the immune status of the host, the route of virus application, and the engineering of the virus with target sites of suitable microRNAs are crucial for use of PD-H as oncolytic virus. This report has some new findings to be reported. However several important issues should be clarified before acceptance to Viruses. The followings are my comments.
Major comments)
1) In Figure 3A, why did untreated mice have smaller size tumor?
2) In right panel of Figure 3D , please indicate the virus titer in pancreas, not RNA, as shown in left panel of Figure 3D.
3) There seems to be discrepancy between Figure 4 A and 4H. Namely, judging from tumor growth curve, treated mice should not show body weight loss between day 18 and 28. What happened in treated mice?
4) In Figure 5D, oncolytic activity of CVB3-PD-375TS and CVB3-PDH should be compared.
5) Between lines 563 and 569 in page 14, both of real-time RT-PCR results and pathological findings should be demonstrated because these data are very important.
Minor comments)
1) The indication of A, B, C and D are missing in Figure 4.
Author Response
Reviewer: #2
Major comments)
1) In Figure 3A, why did untreated mice have smaller size tumor?
Answer: We inoculated two tumors into the both flanks of each mouse. One tumor was treated with PD-H or PBS and other tumor left without any treatment. In Figure 3A, injected and untreated tumors are in the same animals and we selected larger tumors for the treatment. Therefore, untreated or uninjected tumors were relatively in smaller size.
2) In right panel of Figure 3D, please indicate the virus titer in pancreas, not RNA, as shown in left panel of Figure 3D.
Answer: Unfortunately, due to high enzymatic activity of pancreatic tissue lysate, which causes cytotoxicity in HeLa cells (used for plaque assays), we cannot perform plaque assay for pancreas. Therefore, unlike other tissues, the analysis of virus presence in pancreas is only possible with real-time RT-PCR.
3) There seems to be discrepancy between Figure 4 A and 4H. Namely, judging from tumor growth curve, treated mice should not show body weight loss between day 18 and 28. What happened in treated mice?
Answer: We did not find a direct correlation between tumor size and body weight loss. Even if at the later time points PD-H treated animals had large tumor size, we did not observe cachexia in these animals. All PD-H treated animals were sacrificed when the tumor size reached upper limit of 1.8 cm3. We assume that independent to tumor size, animals starts to develop cachexia around 21 days post tumor inoculation (it was shown also in other studies). Presumably, difference in immune cell profile rather than small tumor size in treated animals helped to circumvent severity of cachexia.
4) In Figure 5D, oncolytic activity of CVB3-PD-375TS and CVB3-PDH should be compared.
Answer: Although PD-H and PD-H-375TS show similar growth properties in vitro, from these data we cannot rule out that PD-H may have higher oncolytic activity than PD-H-375TS in vivo. However, PD-H induced severe side effects after intraperitoneal application in healthy Balb/C mice, as we were able to show in our preliminary experiment. We therefore made a conscious decision to develop PD-H-375TS and to use only this virus and not PD-H in the tumor model of intraperitoneal carcinomatosis. Therefore no in vivo data are available for PD-H in this model. According to the safety of PD-H-375TS in the model, we believe that only PD-H-375TS has sufficient safety to potentially be used in a corresponding application in humans.
5) Between lines 563 and 569 in page 14, both of real-time RT-PCR results and pathological findings should be demonstrated because these data are very important.
Answer: We found viral RNA in the pancreas by real-time RT-PCR in only one of six mice with intraabdominal growing CT-26 tumors after treatment with PD-H-375TS, while no viral RNA was found in the pancreas of the other animals. In contrast, virus was not detected in the heart of any of the six animals. In addition, the histological examination showed no signs of pathological changes in the pancreas and heart, confirming the absence of viral infection in both organs. These data show that the insertion of the miR-375TS into the PD-H genome almost completely inhibited virus replication in the most sensitive organs of the murine organism. As described in the text, the titers in the pancreas of one animal were very low (1 x 102 virus genome copies/µg). As suggested by the referee, this data has now been inserted into the text and can be found at page 14, lines 571-573.
Minor comments)
1) The indication of A, B, C and D are missing in Figure 4.
Answer: We appreciate the reviewer’s suggestion, Figure 4 has been corrected.